# Understanding the biology, morbidity and social contexts of adolescent tuberculosis: a prospective observational cohort study protocol (Teen TB)

Jeremi Swanepoel [1,2] Klassina Zimri,[1] Marieke M van der Zalm,[1] Graeme Hoddinott [1] Megan Palmer,[1] Alex Doruyter,[3,4] Gezila De Beer,[1] Leanie Kleynhans,[5] Sarah M Johnson,[6] Vita Jongen,[7] Dillon Wademan [1] Khanyisa Mcimeli,[1] Stephanie Jacobs,[1] Ruan Swanepoel,[8] Gert Van Zyl,[9] Brian W Allwood,[10] Stephanus Malherbe,[5] Charlotte Heuvelings,[1] Stephanie Griffith-Richards,[11] Elizabeth Whittaker,[6] David A J Moore,[2] H Simon Schaaf [1] Anneke C Hesseling,[1] James A Seddon[1,6]

**Correspondence to**
Doctor James A Seddon;
jseddon@sun.ac.za

## ABSTRACT

**Introduction** A considerable burden of the tuberculosis (TB) epidemic is found in adolescents. The reasons for increased susceptibility to TB infection and higher incidence of TB disease in adolescence, compared with the 5–10 years old age group, are incompletely understood. Despite the pressing clinical and public health need to better understand and address adolescent TB, research in this field remains limited.

**Methods and analysis** Teen TB is an ongoing prospective observational cohort study that aims to better understand the biology, morbidity and social context of adolescent TB. The study plans to recruit 50 adolescents (10–19 years old) with newly diagnosed microbiologically confirmed pulmonary TB disease and 50 TB-exposed controls without evidence of TB disease in Cape Town, South Africa, which is highly endemic for TB. At baseline, cases and controls will undergo a detailed clinical evaluation, chest imaging, respiratory function assessments and blood collection for viral coinfections, inflammatory cytokines and pubertal hormone testing. At 2 weeks, 2 months and 12 months, TB disease cases will undergo further chest imaging and additional lung function testing to explore the patterns of respiratory abnormalities. At week 2, cases will complete a multicomponent quantitative questionnaire about psychological and social impacts on their experiences and longitudinal, in-depth qualitative data will be collected from a nested subsample of 20 cases and their families.

**Ethics and dissemination** The study protocol has received ethical approval from the Stellenbosch University Health Research Ethics Committee (N19/10/148). The study findings will be disseminated through peer-reviewed publications, academic conferences and formal presentations to health professionals. Results will also be made available to participants and caregivers.

## STRENGTHS AND LIMITATIONS OF THIS STUDY

⇒ A key strength of this study is the use of multiple imaging methods and respiratory function testing modalities to delineate the patterns of structural and functional respiratory abnormalities in adolescent pulmonary tuberculosis (TB).

⇒ Our longitudinal study design will permit us to characterise the sociological and psychological impact of both drug-susceptible and multidrug-resistant TB on adolescents over time.

⇒ The comprehensive measurement of pubertal hormones, viral coinfections and inflammatory cytokines relevant to TB biology will contribute to identifying host factors that impact the human immune response to *Mycobacterium tuberculosis*.

⇒ The small sample size and susceptibility to loss to follow-up are limitations of this study.

## INTRODUCTION

There are an estimated 1.6 billion adolescents (aged 10–19 years) living in the world today, with 90% in low/middle-income countries (LMICs).[1 2] Modelling studies suggest that 750 000 adolescents develop tuberculosis (TB) disease each year globally,[3] yet adolescent TB remains a neglected field of research. Adolescents have unique biological and social circumstances that differ from both younger children and adults. These include hormonal and immunological pubertal changes, increasing autonomy, extensive socialisation and risk-taking behaviour. These unique features have been poorly studied in

resource-limited settings, particularly in relation to their impact on health.

For reasons incompletely understood, the risk of progression from TB infection to TB disease rises in adolescence.[4] This may be related to the impact of puberty on the immunological response to *Mycobacterium tuberculosis* (*Mtb*),[4] or due to coinfections, nutrition and substance abuse (alcohol, smoking, illicit drugs) impacting on TB pathogenesis, as well as changes in social life impacting on exposure risks. Adolescents tend to develop infectious, cavitary forms of disease,[5–9] thus having a greater impact than younger children on TB transmission. Furthermore, there is extensive social mixing,[3 5] late presentation of TB disease,[10] poor adherence to treatment and follow-up and high rates of loss to follow-up.[11] These may all have a substantial impact on TB within the adolescent population and on propagation of the epidemic.[6] Outcomes for adolescents with TB are worse than most other population groups.[12] Treatment challenges are more complex when multidrug-resistant TB (MDR-TB; defined as disease caused by *Mtb* resistant to at least rifampicin and isoniazid) is present or when there is HIV coinfection.[13] Finally, the impact of TB on respiratory morbidity in adolescents is almost entirely unexplored. Emerging evidence in adults demonstrates that more than half of individuals develop clinically relevant respiratory morbidity post pulmonary TB,[14 15] and there is evolving appreciation that respiratory insults in childhood and adolescence have lifelong consequences.[16 17] Any functional impairment may have an impact on their quality of life, their future educational and employment potential, as well as their long-term health.[18 19] This illustrates the importance of exploring both pulmonary function and psychosocial aspects of adolescent TB in research.

TB has been identified as a key research priority for adolescent health in LMICs,[20] and adolescent TB is a crucial component of the revised WHO Roadmap for ending TB.[21] We therefore set out to conduct a study to examine the biology, morbidity and psychosocial contexts of adolescent TB, named Teen TB. This study will provide a vehicle to explore how these different elements of adolescent TB interact. Teen TB will also allow an unparalleled opportunity to explore the patterns of abnormalities found on chest radiography, ultrasound imaging and [18]F-fluorodeoxyglucose (FDG) positron emission tomography (PET) combined with chest radiography and chest CT in adolescents with TB. In addition, it will enable these findings to be correlated with the extent of respiratory deficit, social contexts and biological markers. Understanding this will inform our understanding of adolescent TB treatment, adherence and transmission and will have implications for future interventional studies.

## Study objectives

The Teen TB study has three main objectives. The first is to evaluate the relationship between baseline lung imaging (advanced and point of care) and respiratory function in adolescents with TB disease. We aim to test the hypothesis that imaging at the time of diagnosis can predict future lung function. The second objective is to explore the psychosocial experience of adolescents affected by drug-susceptible (DS) and MDR-TB and evaluate if the diagnosis of TB is associated with experiences of stigma. The third objective is to explore how pubertal hormones and viral coinfections (notably cytomegalovirus (CMV)) influence the immune response to *Mtb*. We hypothesise that as adolescents pass through puberty, they generate an increasingly pronounced myeloid inflammatory response to *Mtb*.

## METHODS AND ANALYSIS
### Study design and setting

Teen TB is a prospective observational cohort study which plans to recruit 50 adolescents (10–19 years) with newly diagnosed microbiologically confirmed TB disease (including both MDR-TB and DS-TB) in Cape Town, Western Cape, South Africa and follow them over a 12-month period. We will also recruit 50 TB-exposed controls (exposed to both DS-TB and MDR-TB). South Africa's Western Cape province is highly endemic for TB and previous studies have shown a substantial burden of TB among adolescents in this setting.[12] The overall TB incidence rate in the Western Cape was 681/100 000 in 2015 and, in a study from the same province, the incidence of TB in adolescents was found to be 450/100 000.[22 23] In 2020–2021, the estimated HIV incidence rates for adults aged 15–49 years in the Western Cape was 0.51%.[24] This high burden setting is an ideal place to carry out adolescent TB studies to meet research gaps.

### Study participants

Any adolescent with newly diagnosed pulmonary TB bacteriologically confirmed on sputum (Xpert-positive or culture-positive), with or without HIV coinfection and who are within the first 14 days of treatment, are eligible to participate in this study. Any adolescent exposed in their household to a case of infectious pulmonary TB and who has no symptoms of TB will be considered for the TB-exposed control group. Those with TB disease will be excluded from the study if they have extrapulmonary TB without evidence of pulmonary TB. TB-exposed controls will be excluded if they have previously recovered from TB disease or if asymptomatic TB disease is suspected and/or diagnosed by baseline study investigations. Those suspected of having asymptomatic TB disease will be considered for the TB disease group if they meet the inclusion criteria. Both participants with TB disease and TB exposure will be excluded if they have a severe illness or any condition causing the adolescent to be clinically unstable or requiring intensive care treatment, if pregnant or breast feeding, if known to have diabetes mellitus and if HIV testing is declined with no recent (<12 months) HIV test result available (table 1).

### Recruitment process

DS-TB recruitment will be carried out in collaboration with clinical staff in health facilities in the Tygerberg,

**Table 1** Inclusion and exclusion criteria for Teen TB

| | Inclusion | Exclusion |
|---|---|---|
| **Tuberculosis disease** | Any adolescent (10–19 years) who:<br>▶ Has a primary diagnosis of newly diagnosed pulmonary TB bacteriologically confirmed on sputum (Xpert-positive or culture-positive), with or without HIV coinfection<br>▶ Is within the first 14 days since diagnosis and thus 14 days of TB treatment | ▶ Extrapulmonary TB without evidence of pulmonary TB<br>▶ Severe illness or any condition causing the adolescent to be clinically unstable or require intensive care treatment<br>▶ Pregnancy or breast feeding (due to exposure to ionising radiation)<br>▶ Diabetes mellitus (due to the potential confounding effects of hyperglycaemia and insulin levels on $^{18}$F-FDG uptake on PET/CT)<br>▶ Participants declining HIV testing for whom a recent (<12 months) HIV test result is not available |
| **Tuberculosis exposure** | Any adolescent (10–19 years) who:<br>▶ Has been exposed in their household in the last 6 months to a case of infectious pulmonary TB<br>▶ Has no symptoms of TB disease | ▶ Previous TB disease<br>▶ Asymptomatic TB disease<br>▶ Severe illness or any condition causing the adolescent to be clinically unstable or require intensive care treatment<br>▶ Pregnancy or breast feeding (to avoid any confounding in comparison with cases)<br>▶ Diabetes mellitus (to avoid any confounding in comparison with cases)<br>▶ Participants declining HIV testing for whom a recent (<12 months) HIV test result is not available |

$^{18}$F-FDG, 18F-fluorodeoxyglucose; PET, positron emission tomography; TB, tuberculosis.

Mitchell's Plain and Khayelitsha subdistricts of the City of Cape Town Metropole. As adolescents present to health facilities in these subdistricts with TB symptoms, we will approach them and sensitise them to the study. If they are diagnosed with DS-TB and meet the eligibility criteria, we will approach them prior to starting TB treatment to see if they would be willing to participate. If willing to participate, informed written consent will be obtained from parents/legal guardians for those under 18 years with additional assent from the adolescent, whereas 18-year and 19-year olds can provide consent for themselves. Initial consultations will take place at their homes, local clinics or at Desmond Tutu TB Centre (DTTC) field sites, as determined to be most convenient to the participant and their family. Given the necessity for all participants to undertake lung function testing at Tygerberg Hospital and to undergo $^{18}$F-FDG PET-CT imaging for those with TB disease, the follow-up visit will be at the iKamva Unit on the Tygerberg Campus of Stellenbosch University (SU).

MDR-TB recruitment will be carried out using the established tuberculosis child multidrug-resistant preventive therapy (TB-CHAMP) trial platform to identify adolescents aged 10–19 years in the Metropolitan area who are newly diagnosed with MDR-TB disease. TB-CHAMP is an ongoing MDR-TB prevention trial that randomises children who have been exposed to MDR-TB in their household to levofloxacin or placebo, with three South African sites, and in collaboration with the MRC Clinical Trials Unit at University College London. The TB-CHAMP trial team identifies every MDR-TB patient in the Cape Town Metropolitan area through laboratory and clinical sources. This is an established platform by which adolescent MDR-TB cases can be identified. We will then, using our established links with routine services, and in collaboration with their healthcare providers, contact these adolescents, following sensitisation to the study by local clinics, ideally prior to starting MDR-TB treatment, and invite them to join the study.

TB-exposed controls are recruited from the same clinics where individuals with TB disease are recruited, and the contacts of the enrolled cases will also be eligible for inclusion. TB-exposed controls must have been exposed to an infectious DS-TB case within the last 6 months. We will again use the TB-CHAMP platform to aid in identifying adolescents who have been exposed to an infectious pulmonary MDR-TB case in the household within the last 6 months.

### Study procedures
The study visit schedule for participants is outlined in table 2 and provides a description of the study procedures and investigations planned for each visit.

### Clinical data collection
At each study visit, we will collect a full clinical TB history, comprehensive clinical data and socioeconomic data from all participants. A clinical examination, including a cardiorespiratory and abdominal exam, will take place at each visit with Tanner staging done at recruitment. A baseline weight and, in those with TB disease, a follow-up weight at 2 weeks, 2 months and 12 months will be recorded. Programmatic and laboratory data (including

**Table 2** Description of study visits and investigations in Teen TB

| Time point | Activity | TB disease | TB exposure |
|---|---|---|---|
| | | n=50 | n=50 |
| Baseline | Consent and recruitment | ● | ● |
| | Complete CRFs | ● | ● |
| | Portable spirometry | ● | ● |
| | Oscillometry | ● | ● |
| | Venepuncture | ● | ● |
| | Chest X-ray | ● | ● |
| | 6-min walk test | ● | ● |
| | St. George's Respiratory Questionnaire | ● | ● |
| | Full laboratory lung function | | ● |
| 2 weeks | Complete CRFs | ● | |
| | Ultrasound | ● | |
| | Portable spirometry | ● | |
| | Oscillometry | ● | |
| | $^{18}$F-FDG PET-CT | ● | |
| | Psychosocial questionnaire | ● | |
| 2 months | Complete CRFs | ● | |
| | Venepuncture | ● | |
| | Chest X-ray | ● | |
| | Ultrasound | ● | |
| | Full laboratory lung function | ● | |
| | Portable spirometry | ● | |
| | Oscillometry | ● | |
| | 6-min walk test | ● | |
| | St. George's Respiratory Questionnaire | ● | |
| 12 months | Complete CRFs | ● | |
| | Oscillometry | ● | |
| | Full laboratory lung function | ● | |
| | Chest X-ray | ● | |
| | 6-min walk test | ● | |
| | St. George's Respiratory Questionnaire | ● | |

CRF, case report form; $^{18}$F-FDG PET-CT, fluorine-18 fluorodeoxyglucose positron emission tomography combined with (X-ray) CT.

microbiology) will be collected from the treating clinic and from the National Health Laboratory Service.

### Blood sample collection

At baseline, all adolescents will be offered an HIV test. If they have not had a test in the last 12 months, blood will be drawn to perform an HIV ELISA. If their HIV status is available from a recent result (from the past 12 months), then this result will be used. Any participant with a new HIV infection diagnosed during this study will be counselled appropriately by a trained member of the research team and referred to routine services for evaluation and initiation of antiretroviral treatment. Participants with TB disease and TB-exposed participants will also have blood collected at baseline for CMV quantitative IgG, RNA sequencing, hormone profiling (including dehydroepiandrosterone sulfate, leptin, testosterone, oestradiol and progesterone levels) and for a panel of inflammatory cytokines relevant to TB biology, selected from previous work by the study team and from the literature. In TB-exposed individuals, blood will be taken for an interferon-gamma release assay. Participants with TB disease will undergo an additional blood draw at 2 months and samples will be stored for later analysis.

### Imaging

All adolescents with TB disease will undergo chest radiography (posteroanterior and lateral) as part of the standard routine care at baseline and will also undergo further chest X-rays (CXRs) at 2 months and 12 months.

Those exposed to TB will have a baseline CXR only. Individuals with TB disease will undergo ultrasound at 2 weeks and 2 months, performed by trained medical personnel and using standard operating procedures. Ultrasound examination will include an evaluation of the chest, the abdomen and the mediastinum (through the suprasternal notch) and will be performed using two portable ultrasound machines—the Butterfly IQ and the Mindray DP 10. Two weeks into treatment, $^{18}$F-FDG PET-CT imaging will be performed for all adolescents with TB disease at the NuMeRI Node for Infection Imaging of SU. This will be carried out using a Vereos Digital PET 64-channel CT scanner (Philips Medical Systems) with a dose-sparing acquisition protocol. Participants will be fasted for 4–6 hours prior to the administration of FDG, which will be administered via an intravenous cannula, using universal precautions and aseptic techniques.

## Respiratory function assessment

Forced expiratory volume in 1 second (FEV1), forced vital capacity (FVC) and the FEV1/FVC ratio will be carried out using a hand-held NDD EasyOne PC (Switzerland) spirometer at baseline in all adolescents and again at 2 weeks and 2 months in those with TB disease. A minimum of three trials up to a maximum of eight attempts will be carried out per participant and at least three acceptable, reproducible best quality curves will be selected. The best value of three will be reported. Four hundred micrograms of salbutamol will be administered via spacer and testing will be repeated after 15 min to assess reversibility. Criteria for acceptability and repeatability of the spirometry curves will be based on European Respiratory Society (ERS)/American Thoracic Society (ATS). Global Lung Initiative reference ranges will be used using the category 'other' to reflect the South African population.[25 26]

The portable THORASYS tremoFlo C-100 Airwave Oscillometry System will be used for additional assessment of pulmonary function. The TremoFlo is a tidal breathing technique that measures airway resistance and reactance and allows for detailed insights into small airway disease. All participants will have oscillometry at baseline, while those with TB disease will have this repeated at 2 weeks, 2 months and 12 months. Oscillometry will be done pre-bronchodilation and post-bronchodilation as described for spirometry.

In collaboration with the Tygerberg Lung Function Laboratory, all individuals with TB exposure but without disease will undergo a full lung function assessment as soon after recruitment as possible. For those with TB disease, the evaluation will take place at 2 months and 12 months post recruitment. This will include spirometry with pre-bronchodilator and post-bronchodilator measurement, diffusion capacity and plethysmography. These tests will be conducted according to ERS/ATS guidelines and local hospital protocols.

All participants will undergo a 6-min walk test (6MWT). A trained technician will carry out the standard protocol as per the ATS guidelines.[27] The results will be compared

with population-based standards.[28] Those with TB disease will undergo this again at 2 months and 12 months. Participants will also complete the St. George's Respiratory Questionnaire (SGRQ) which assesses self-reported symptoms of breathlessness, and impact of this on activities and psychosocial function. Those with TB disease will have this repeated at 2 months and 12 months.

## Psychosocial context data

All individuals with TB disease will complete a multicomponent quantitative questionnaire (see online supplemental file 1) about psychological and social impacts on their experiences (including stigma, psychological wellbeing, social support and similar) at week 2. These data will be analysed using common descriptive statistics. We will also collect longitudinal in-depth qualitative data (3–4 interactions over the course of their participation in the project) with a nested, subsample of 20 adolescents, purposively sampled for diversity and richness, stratified 1:1 among adolescents with DS-TB and MDR-TB. The life course experiences of this subgroup will be characterised, including familial/peer social contexts, experiences of TB disease including disability, sexual experiences and time of sexual debut, HIV, smoking, substance abuse, violence, processes of peer support and socialisation, impact on education, episodic and chronic threats to mental wellbeing, accessing health services, TB treatment adherence and quality of life. We will use ATLAS.ti to conduct thematic and case descriptive analyses.

## Statistical analysis
### Power and sample size

While multiple different analyses are planned for the different components of the study (clinical, immunology, radiology, social science, virology), we calculated the sample size based on ability to detect differences in lung function. Even within this discipline, several lung function parameters will be measured during this study, but sample size calculations were performed based on assumptions related to spirometry. A reduced FVC, compared with age and height reference standards, indicates restrictive lung disease, whereas a reduced FEV1 and FEV1/FVC ratio, compared with reference standards, indicate obstructive lung disease. Both patterns of lung disease have been seen in adults following pulmonary TB. To demonstrate a difference between healthy individuals in our context (assumed to have a FEV1/FVC ratio of 100%; SD: 15%), and adolescents with pulmonary TB disease (assumed to have a ratio of 90%), with 90% power and an alpha of 0.05, we would need 47 adolescents with pulmonary TB and 47 healthy controls. Therefore, we plan to recruit 50 cases and 50 controls given the uncertainty in these estimates and to allow for any lost data or incomplete evaluations.

### Assessment of study endpoints/primary objectives

Descriptive analysis will be used to characterise the study populations, to compare the TB case and TB-exposed

groups and aid in identifying differences between groups with respect to potential confounders. Categorical variables will be expressed as percentages and continuous variables will be expressed as mean±one SD or median with IQR, depending on the distribution of the data.

In those with TB disease, we will categorise disease on CXR as severe and non-severe, using established criteria[29] and will compare mean z-score of FVC, FEV1 and FEV1/FVC ratio at 2 months and 12 months in those with severe and non-severe disease using a paired sample t-test. We will also evaluate the relationship between severe and non-severe disease on CXR with other measures of functional tests (oscillometry, diffusion capacity, plethysmography, 6MWT and SGRQ). We will perform similar analyses evaluating the ability of ultrasound, CT and [18]F-FDG PET-CT to predict different measures of lung function at 2 months and 12 months. We will compare the lung function of adolescents with TB disease and those with TB exposure using multivariable logistic regression to adjust for potential confounders and to determine the impact of TB on lung health. From a radiology perspective, we will evaluate the ability of CXR and lung ultrasound to identify findings as seen on CT chest (considered as the imaging 'gold standard' for this analysis) using either contingency tests or receiver operating characteristic (ROC) analysis, where appropriate.

Further multivariable analyses will be performed to explore the impact of pubertal hormones on inflammatory cytokines and how the inflammatory cytokines of adolescents with TB disease and TB exposure differ. Immune responses to viral coinfections in participants with TB disease and those with TB exposure will also be compared using regression analysis. Analyses will be performed using STATA (V.17 STATA Corp.).

### Data management and confidentiality

At recruitment, each individual will be allocated a unique study number. Personal identifier details will be recorded (eg, name, address and telephone number) but these will thereafter be kept securely and will remain separate from any clinical/epidemiological data, laboratory samples or data resulting from laboratory analysis. These research data will only be linked to the study number. All qualitative data will be audio recorded and transcripts of these recordings anonymised prior to presentation for any audiences outside of the study team. All data (clinical, epidemiological psychosocial and qualitative) will be stored on a secure database at DTTC.

### Patient and public involvement

The development of the protocol, the ethics submission and the strategy for implementation of the study were undertaken in close collaboration with the DTTC Community Advisory Board (CAB). The CAB assists with building and fostering partnerships between researchers and local study communities impacted by TB. The CAB will provide feedback to the research team regarding local issues at clinical research sites and will inform the team of any community concerns that can affect the conduct and implementation of study procedures.

During participant follow-up visits, feedback from adolescents and their caregivers will be used to assess the perceived burden of study interventions and to identify any aspects of their involvement that require improvement or adaptation.

### ETHICS AND DISSEMINATION

The study protocol was approved by the SU Health Research Ethics Committee on 25 February 2020 (N19/10/148). The study will be conducted according to South African and internationally accepted ethical standards, including the Declaration of Helsinki and the South African Good Clinical Practice guidelines. For recruitment, the parents/legal guardians/participants will be required to provide written informed consent in their first language. Adolescents and families will be informed that they are under no obligation to enter the study and that they can withdraw at any time during the study, without having to give a reason.

The study findings will be disseminated through peer-reviewed publications, academic conferences and formal presentations to health professionals and practitioners. Findings will also be communicated to study participants and their caregivers.

### PROJECT STATUS

Recruitment to the study started in November 2020. Follow-up of participants is still ongoing, and the final visit of the last participant with TB disease is planned to be in July 2022. The current protocol is version 1.0.

**Author affiliations**
[1]Desmond Tutu TB Centre, Department of Paediatrics and Child Health, Faculty of Medicine and Health Sciences, Stellenbosch University, Cape Town, South Africa
[2]Faculty of Infectious and Tropical Diseases, London School of Hygiene and Tropical Medicine, London, UK
[3]NuMeRI Node for Infection Imaging, Central Analytical Facilities, Stellenbosch University, Cape Town, South Africa
[4]Division of Nuclear Medicine, Department of Medical Imaging and Clinical Oncology, Stellenbosch University, Cape Town, South Africa
[5]South African Medical Research Council Centre for Tuberculosis Research, Division of Molecular Biology and Human Genetics, Department of Biomedical Sciences, Faculty of Medicine and Health Sciences, DSI-NRF Centre of Excellence, Stellenbosch University, Cape Town, Western Cape, South Africa
[6]Department of Infectious Disease, Imperial College London, London, UK
[7]Department of Infectious Diseases, Public Health Service of Amsterdam, Amsterdam, Netherlands
[8]Division of Pulmonology, Department of Internal Medicine, Tygerberg Hospital, Cape Town, Western Cape, South Africa
[9]Division of Medical Virology, Department of Pathology, Stellenbosch University Faculty of Medicine and Health Sciences, Cape Town, South Africa
[10]Division of Pulmonology, Department of Medicine, Faculty of Medicine and Health Sciences, Stellenbosch University, Cape Town, Western Cape, South Africa
[11]Division of Radiodiagnosis, Department of Medical Imaging and Clinical Oncology, Faculty of Medicine and Health Sciences, Stellenbosch University, Cape Town, Western Cape, South Africa

**Collaborators** Not applicable.

**Contributors** The study concept and design were conceived by J Seddon. J Seddon is also the principal investigator for the Teen TB study. MvdZ, GH, MP, AD, S Johnson, GvZ, BA, EW, DM, SS and AH provided additional methodological and/or clinical advice. KZ, GdB, KM, S Jacobs, RS, CH and SG-R will conduct screening and data collection. DW assisted with psychosocial questionnaire development. Analyses will be performed by VJ, LK, CH and SM. J Swanepoel prepared the first draft of the manuscript and will assist with data analysis. All authors provided edits and critiqued the manuscript.

**Funding** J Seddon is supported by a Clinician Scientist Fellowship jointly funded by the UK Medical Research Council (MRC) and the UK Department for International Development (DFID) under the MRC/DFID Concordat agreement (MR/R007942/1).

**Competing interests** None declared.

**Patient and public involvement** Patients and/or the public were involved in the design, or conduct, or reporting, or dissemination plans of this research. Refer to the Methods section for further details.

**Patient consent for publication** Not applicable.

**Provenance and peer review** Not commissioned; externally peer reviewed.

**ORCID iDs**
Jeremi Swanepoel http://orcid.org/0000-0002-9338-2911
Graeme Hoddinott http://orcid.org/0000-0001-5915-8126
Dillon Wademan http://orcid.org/0000-0003-2222-7401
H Simon Schaaf http://orcid.org/0000-0001-5755-4133

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
