## [Reviewer comments · BMJ Open]

ARTICLE DETAILS

TITLE (PROVISIONAL)	Understanding the biology, morbidity and social contexts of adolescent tuberculosis: a prospective observational cohort study protocol (Teen TB)
AUTHORS	Swanepoel, Jeremi; Zimri, Klassina; van der Zalm, Marieke; Hoddinott, Graeme; Palmer, Megan; Doruyter, Alex; De Beer, Gezila; Kleynhans, Leanie; Johnson, Sarah; Jongen, Vita; Wademan, Dillon; Mcimeli, Khanyisa; Jacobs, Stephanie; Swanepoel, Ruan; U Van Zyl, Gert; Allwood, Brian; Malherbe, Stephanus; Heuvelings, Charlotte; Griffith-Richards, Stephanie; Whittaker, Elizabeth; Moore, Dave; Schaaf, Simon; Hesselings, A C; Seddon, James

VERSION 1 – REVIEW

REVIEWER	Walker, Timothy
REVIEW RETURNED	06-Jun-2022

GENERAL COMMENTS	This is a potentially interesting study and I look forward to reading the results. I have a few questions:  1. Is asymptomatic TB (CXR findings) be an exclusion criteria for the TB exposure arm? It's currently not clear from the protocol. 2. Why are you not following up the TB exposure patients beyond the first encounter, given that some of those might develop disease within the study period? 3. What determines the timing of the ultra sound at 2 weeks, 2 months, but not at 12 months? 4. Do you plan to recruit a certain number of patients with DS and MDR disease, or just leave the proportion to chance? Same question for the exposed individuals - how many people exposed to DS and how many exposed to MDR are you seeking to recruit?
--

REVIEWER	Hashmi, S. Shahrukh University of Texas Health Science Center Houston
REVIEW RETURNED	13-Aug-2022

GENERAL COMMENTS	This is a review of a study protocol for the Teen TB research project. It is a prospective observational cohort study to understand the biology, morbidity and social contexts of adolescent TB. Overall, this is a well designed study with a clearly written protocol. However, some questions were raised during this review: Methods And Analysis section - "Study design and setting" sub-section: If known, please add info on TB incidence rate among adolescents. Currently, overall
--

prevalence is given. Similarly, only antenatal HIV prevalence is provided. If known, please provide adolescent rates.

Recruitment Process section -

Will there be / was there age-based block or stratified sampling in the recruitment of the 50 TB-diseased adolescents? I'm wondering what the age distribution for TB is in the study population. If it is severely skewed (e.g. older adolescents), would the study be able to recruit enough younger adolescents to identify any age-related differences in the natural history of TB in adolescents? As recruitment into the study is nearly complete, the sampling cannot be altered now. However, if there was any age-based sampling criteria, please mention that. If not and there was a severe skewed age distribution (e.g. vast majority aged 16+), then it should be noted during eventual publication of the results and how it impacts any generalizability with respect to all adolescents.

Study Procedures section -

In the "Respiratory function assessment" subsection, it states that "all individuals with TB exposure but without disease will undergo a full lung function assessment two to four weeks after recruitment". Does this mean that the TB exposure (control) group individuals will be assessed for symptomatic TB (i.e. TB disease) again at the 2-4 week period prior to their full lung function test? If symptomatic will they be excluded from the study (as they no longer fit the TB exposure I/E criteria)? Please clarify if they will be re-assessed for symptomatic TB at the time of the full lung function tests at 2-4 weeks.

Statistical Analysis section -

There is longitudinal information collected on the TB-diseased cases and it is stated that lung function tests (as z-scores) between severe and non-severe cases will be compared at 2 and 12 months using a paired t-test. Are there plans for multivariable analyses to adjust for potential confounders or effect modifiers? Either repeated measures ANOVA models, or generalized linear mixed models or other stratified analyses? The only test listed in the protocol is the paired t-test and although other comparisons are mentioned, it is not clear if these are all done independent of each other or which specific analytic tools will be utilized. For example, it states "From a radiology perspective, we will evaluate the ability of CXR and lung ultrasound to identify findings as seen on CT chest" - I'm assuming this will involve categorical variables and utilize contingency tests, but that is not clear. Or do the authors plan on performing a ROC (Receiver Operating Characteristic) analysis to assess the ability of CXR and lung U/S? This section may benefit from some more detail.

General Comment

The section on recruitment processes states that the "contacts of the enrolled cases will also be eligible for inclusion" as TB-exposed controls. I'm assuming this means that two adolescents in the same household, potentially siblings, may be recruited into the study - one as a case and the other as a control. Does this apply to the recruitment of TB-diagnosed cases as well, where two siblings who both have TB-disease can both be recruited as cases

	into the study? What impact would this "familial correlation" have on the lung function tests and the sample size calculations and will it limit the interpretation from this study? Will there be any adjustment for familial or household aggregation in the eventual analytic plan not only for the lung function tests but also the psychosocial assessments planned or the evaluation of other viral co-morbidities? I'm not saying there needs to be and do not know how commonly these scenarios have arisen during the recruitment process, but if they have and there are plans to adjust during the analysis it should be stated briefly.
--	--

VERSION 1 – AUTHOR RESPONSE

Reviewer: 1
 Timothy Walker

Comment 1:
 Is asymptomatic TB (CXR findings) an exclusion criteria for the TB exposure arm? It's currently not clear from the protocol.

Response 1:
 We thank the reviewer for the valuable question. Asymptomatic TB disease was treated as an exclusion criteria for the TB exposure arm but this was not clearly stated in the manuscript. Once an adolescent has had the initial imaging investigations and TB disease is suspected and/or diagnosed they will be referred to appropriate care and considered for inclusion into the disease arm. We have amended the main document to include this TB exposure arm exclusion criteria.

Comment 2:
 Why are you not following up the TB exposure patients beyond the first encounter, given that some of those might develop disease within the study period?

Response 2:
 The need to follow up the adolescents with TB exposure depends on the aim the study and the questions that we plan to answer. For Teen TB, we are not planning to evaluate the risk of progression to TB disease which would require a different study design, regular follow-up of those with TB exposure and a larger sample size. We only required baseline findings from healthy controls for comparative purposes.

Comment 3:
 What determines the timing of the ultrasound at 2 weeks, 2 months, but not at 12 months?

Response 3:
 We did not carry out an ultrasound at baseline due to infection control concerns relating to the duration of the scan and the intensity of the interaction with the sonographer. At 2 weeks, participants would have completed at least two weeks of anti-TB treatment and we felt it was then safe to undertake the lung ultrasound. The primary questions that we wanted to address were: 1) how does chest ultrasound compare to other imaging modalities in adolescent with TB disease? and 2) do lung ultrasound findings change over time in adolescents treated for TB disease? To answer these questions it was sufficient to carry out ultrasounds at 2 weeks and 2 months. Undertaking an ultrasound in our context requires quite a time commitment from both the study team and the participant and we felt that as it was not required at 12 months to answer our primary research questions, we elected to not carry it out then.

Comment 4:

Do you plan to recruit a certain number of patients with DS and MDR disease, or just leave the proportion to chance? Same question for the exposed individuals - how many people exposed to DS and how many exposed to MDR are you seeking to recruit?

Response 4:

We hypothesise that there are minimal biological differences between adolescents with MDR-TB disease and DS-TB disease. However, the experiences that are faced by adolescents with MDR-TB are substantially different, in terms of interaction with health services, relationships with peers, family and impact on jobs and education. It also is a far more stigmatising condition. We therefore wanted to recruit sufficient individuals with MDR-TB so that these perspectives were captured in the qualitative work. We sought to include at least 10 individuals with MDR-TB so that half of the qualitative sub-study (total of n=20) would have MDR-TB. Although we did not have a fixed target proportion for DS-TB vs. MDR-TB, we aimed for roughly a ratio of 4:1 DS-TB: MDR-TB to capture these perspectives. We replicated these proportions for the healthy controls.

Reviewer: 2

Dr. S. Shahrukh Hashmi, University of Texas Health Science Center Houston

Comment 1:

Methods And Analysis section -

"Study design and setting" sub-section: If known, please add info on TB incidence rate among adolescents. Currently, overall prevalence is given. Similarly, only antenatal HIV prevalence is provided. If known, please provide adolescent rates.

Response 1:

Thank you for the comment. We have included the TB incidence rate among adolescents in the Western Cape as well as the HIV incidence rate for adults aged 15-49 years (based on the Thembisa Model) in the same province.

Comment 2:

Recruitment Process section -

Will there be / was there age-based block or stratified sampling in the recruitment of the 50 TB-diseased adolescents? I'm wondering what the age distribution for TB is in the study population. If it is severely skewed (e.g. older adolescents), would the study be able to recruit enough younger adolescents to identify any age-related differences in the natural history of TB in adolescents? As recruitment into the study is nearly complete, the sampling cannot be altered now. However, if there was any age-based sampling criteria, please mention that. If not and there was a severe skewed age distribution (e.g. vast majority aged 16+), then it should be noted during eventual publication of the results and how it impacts any generalizability with respect to all adolescents.

Response 2:

We did not use any age-based sampling criteria. Time restrictions and resource constraints limited our ability to use any age-based block or stratified sampling in the recruitment of TB disease adolescents. If an age skew is identified then we will adjust for this in our analyses and it will be noted in the eventual publication. The impact of an age skew on the generalisability of our findings will also be noted during eventual publication.

Comment 3:

Study Procedures section -

In the "Respiratory function assessment" subsection, it states that "all individuals with TB exposure but without disease will undergo a full lung function assessment two to four weeks after recruitment". Does this mean that the TB exposure (control) group individuals will be assessed for symptomatic TB (i.e. TB disease) again at the 2-4 week period prior to their full lung function test? If symptomatic will they be excluded from the study (as they no longer fit the TB exposure I/E criteria)? Please clarify if they will be re-assessed for symptomatic TB at the time of the full lung function tests at 2-4 weeks.

Response 3:

We thank the reviewer for the question and agree that more clarification is needed. When we drafted the protocol we were concerned that due to logistic constraints it may not have been possible to carry out all the lung function assessments soon after recruitment, as had been intended. The lung function testing is performed by a respiratory clinical technologist at Tygerberg Academic Hospital and the amount of testing slots per week are limited. However, during recruitment of participants we have managed to carry out lung function in the couple of days follow recruitment in the healthy controls. We have therefore adjusted the text to read:

In collaboration with the Tygerberg Lung Function Laboratory, all individuals with TB exposure but without disease will undergo a full lung function assessment as soon after recruitment as possible.

Comment 4:

Statistical Analysis section -

There is longitudinal information collected on the TB-diseased cases and it is stated that lung function tests (as z-scores) between severe and non-severe cases will be compared at 2 and 12 months using a paired t-test. Are there plans for multivariable analyses to adjust for potential confounders or effect modifiers? Either repeated measures ANOVA models, or generalized linear mixed models or other stratified analyses? The only test listed in the protocol is the paired t-test and although other comparisons are mentioned, it is not clear if these are all done independent of each other or which specific analytic tools will be utilized. For example, it states "From a radiology perspective, we will evaluate the ability of CXR and lung ultrasound to identify findings as seen on CT chest" - I'm assuming this will involve categorical variables and utilize contingency tests, but that is not clear. Or do the authors plan on performing a ROC (Receiver Operating Characteristic) analysis to assess the ability of CXR and lung U/S? This section may benefit from some more detail.

Response 4:

Thank you for the comment. Teen TB serves as vehicle to explore the biology, morbidity and social contexts of adolescent TB; therefore, multiple analyses will be performed by researchers involved with different aspects of the project. We did not feel it was necessary to mention or outline all planned analyses. However, we have added some more detail to the 'Assessment of study endpoints/ primary objectives' section.

Comment 5:

General Comment

The section on recruitment processes states that the "contacts of the enrolled cases will also be eligible for inclusion" as TB-exposed controls. I'm assuming this means that two adolescents in the same household, potentially siblings, may be recruited into the study - one as a case and the other as a control. Does this apply to the recruitment of TB-diagnosed cases as well, where two siblings who both have TB-disease can both be recruited as cases into the study? What impact would this "familial correlation" have on the lung function tests and the sample size calculations and will it limit the interpretation from this study? Will there be any adjustment for familial or household aggregation in the eventual analytic plan not only for the lung function tests but also the psychosocial assessments planned or the evaluation of other viral co-morbidities? I'm not saying there needs to be and do not

know how commonly these scenarios have arisen during the recruitment process, but if they have and there are plans to adjust during the analysis it should be stated briefly.

Response 5:

Thank you for the valuable comment. Most controls were same household contacts of enrolled cases (sibling or other family member). We only had one instance where two siblings from the same household, who both had TB disease, were recruited into the study. The reviewer makes a very valid point by stating that this “familial correlation’ may impact respiratory function assessment and the evaluation of viral co-infections. Therefore, familial or household clustering will be adjusted for in all our analyses. Teen TB is a relatively small study that will generate hypothesis which will be explored in a subsequent larger study.

VERSION 2 – REVIEW

REVIEWER	Hashmi, S. Shahrukh University of Texas Health Science Center Houston
REVIEW RETURNED	01-Nov-2022

GENERAL COMMENTS	The authors have adequately responded to the reviewers' comments and stipulations. As this is a protocol submission rather than a submission of an original research, I feel that the manuscript as written is adequate. There are still some issues with lack of detail but a lot of those are dependent on logistical and other issues that may be particular to the study site. The lack of that information does not preclude others from replicating this study as site specific modifications may be warranted for any future study building off of the current protocol. Other issues that were raised can be dealt with in a future publication that reports on the results of this study and, in my opinion, are not necessary for inclusion in this submission.
--